# Germline variation contributes to false negatives in CRISPR-based experiments with varying burden across ancestries

Sean A. Misek[1,2,3], Aaron Fultineer[1], Jeremie Kalfon [1], Javad Noorbakhsh[1], Isabella Boyle [1], Priyanka Roy[1], Joshua Dempster [1], Lia Petronio[1], Katherine Huang[1], Alham Saadat[1], Thomas Green[1], Adam Brown[1], John G. Doench [1], David E. Root [1], James M. McFarland [1], Rameen Beroukhim [1,2,4] ✉ & Jesse S. Boehm [1,3,4] ✉

Reducing disparities is vital for equitable access to precision treatments in cancer. Socioenvironmental factors are a major driver of disparities, but differences in genetic variation likely also contribute. The impact of genetic ancestry on prioritization of cancer targets in drug discovery pipelines has not been systematically explored due to the absence of pre-clinical data at the appropriate scale. Here, we analyze data from 611 genome-scale CRISPR/Cas9 viability experiments in human cell line models to identify ancestry-associated genetic dependencies essential for cell survival. Surprisingly, we find that most putative associations between ancestry and dependency arise from artifacts related to germline variants. Our analysis suggests that for 1.2-2.5% of guides, germline variants in sgRNA targeting sequences reduce cutting by the CRISPR/Cas9 nuclease, disproportionately affecting cell models derived from individuals of recent African descent. We propose three approaches to mitigate this experimental bias, enabling the scientific community to address these disparities.

CRISPR is an increasingly important tool in biomedical research[1] and can be leveraged to identify genetic determinants of a range of phenotypes, including cancer cell fitness[2]. CRISPR-mediated genome editing requires homology between a guide RNA and a genomic locus; mismatches between these two sequences are expected to interfere with editing efficiency[3]. While 99.9% of the human genome is identical from one person to the next, the remaining 0.1% includes variants that influence physical characteristics and health. The degree to which the consensus reference genome used for guide design fully captures human germline and ancestral diversity is improving[4]. However, current CRISPR/Cas9 libraries were designed from reference genomes that preceded the Pangenome reference and did not capture the diversity of human variation[3,5].

Recent advances in leveraging systematic CRISPR/Cas9 experiments to map genes that are required for the survival of cancer cells (herein referred to as cancer dependencies or gene dependencies) across hundreds of cellular models provide new opportunities to understand cancer targets and the molecular features of cancers that drive sensitivity to response[6,7]. The Cancer Dependency Map[8] represents the largest such resource and currently includes data from genome-scale CRISPR/Cas9 gene essentiality screens across 1070 cancer cell lines reflecting 31 cancer lineages to detect essential genes and their relationships with predictive molecular biomarkers. These data have led to the discovery of multiple dependency-associated somatic alterations including dependency on *WRN* in cell lines with microsatellite instability[9] and dependency on *PRMT5* in cells with

[1]Broad Institute of MIT and Harvard, Cambridge, MA 02142, USA. [2]Departments of Cancer Biology and Medical Oncology, Dana-Farber Cancer Institute, Boston, MA 02215, USA. [3]Koch Institute, Massachusetts Institute of Technology, Cambridge, MA 02142, USA. [4]These authors jointly supervised this work: Rameen Beroukhim, Jesse S. Boehm. ✉e-mail: Rameen_Beroukhim@dfci.harvard.edu; boehm@mit.edu

genomic *MTAP* deletions[10], amongst others. While the role of germline variation has been demonstrated to contribute to drug sensitivity[11], the degree to which ancestry and germline variation contributes to gene dependencies has not been systematically interrogated. In addition, germline variation can cause mismatches between CRISPR guides and the genome of a given cellular model, confounding associations between germline genetics and cellular dependencies.

Here, we systematically interrogate the degree to which genetic ancestry and germline variation contributes to cancer dependencies, and the extent to which this is due to sgRNA mismatches with germline genotypes. A majority of CRISPR/Cas9 guides in genome-scale libraries are affected by this artifact, which disproportionately affects individuals of recent African descent. We demonstrate the impact of this experimental artifact on identification of genetic ancestry-associated dependencies and highlight putative ancestry-dependency associations not resulting from this artifact. Finally, we highlight three approaches to mitigate the impact of this experimental artifact in CRISPR guide design.

## Results

### Identification of ancestry-associated genetic dependencies

We started by analyzing putative ancestry-associated cancer dependencies using data from The Cancer Dependency Map (Fig. 1a). First,

we evaluated cell line genetic ancestry, considering the possibility of ancestry admixture. While previous reports have evaluated cell line genetic ancestry at genome scale[12–16], we hypothesized that such global assessments may preclude the discovery of regional germline associations with dependencies. We therefore systematically cataloged local ancestral haplotypes across the genomes of the 994 (out of 1829 total) cell line models in the Cancer Cell Line Encyclopedia collection for which publicly available Affymetrix SNP6 germline variant data have been analyzed[14], leveraging germline variants from 10,345,968 SNPs genome-wide to infer local ancestry. Specifically, we divided the genome into blocks comprising 0.2 centimorgans (with a median of 580 SNPs per block) and characterized each block as deriving from one (homozygous) or two (heterozygous) of five major continental genetic ancestry groups: African (AFR), American (AMR), East Asian (EAS), European (EUR), and South Asian (SAS) (Fig. 1b). In admixed individuals, individual blocks might derive from two of these ancestries, reflecting both maternal and paternal contributions.

At a global level, our results support previous observations[14,16] that existing cell lines are overwhelmingly derived from individuals of either EUR or EAS ancestry. We assigned a predominant ancestry to cell lines that derived over 80% of their DNA from that ancestry group and called those without a predominant ancestry Admixed. Of the 994 cell lines profiled in this study, over 90% of them are predominantly EUR

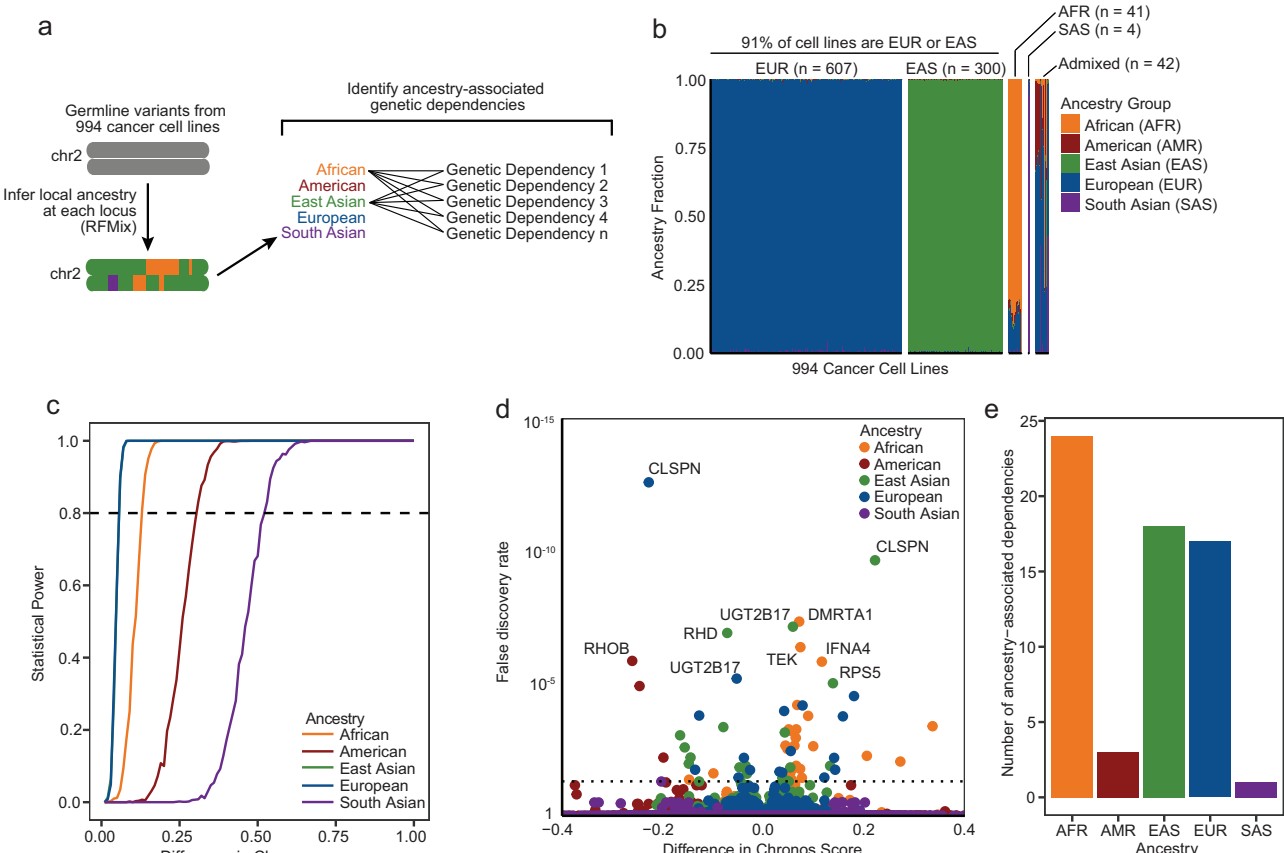

**Fig. 1 | Identification of ancestry-associated genetic dependencies. a** Schematic describing the methodology for identifying ancestry-associated genetic dependencies. **b** Local ancestry assignments for 994 cancer cell lines. The genomic fraction corresponding to one of five major continental ancestry groups (African, American, East Asian, European, or South Asian) is indicated. Cell lines whose genome is comprised >80% of a single ancestry group are denoted as being of the given ancestry group (African [*n* = 41], American [*n* = 0], East Asian [*n* = 300], European [*n* = 607], South Asian [*n* = 4]), otherwise the cell line is denoted as being Admixed [*n* = 42]. Raw data are described in Source Data 1B. **c** Statistical power for detecting ancestry-associated dependencies for each ancestry group. Raw data are

described in Source Data 1C. **d** Associations between ancestry and Chronos scores. For each gene, cell lines were binned by local ancestry at the transcription start site. The association between ancestry and dependency on the gene in question was computed with linear regression with correction for cancer lineage as a covariate. Associations with false discovery rate (FDR) q-values of <0.05 (dashed line) were called as significant association between ancestry and dependency on the gene in question. Raw data are described in Source Data 1D. **e** Total numbers of putatively significant ancestry-associated genetic dependencies for each major ancestry group. AFR African, AMR American, EAS East Asian, EUR European, SAS South Asian. Raw data are described in Source Data 1E.

(61%) or EAS (30%) (Fig. 1b, Source Data 1B). Only 41 (4%) of the cell lines were predominantly AFR. When taking local ancestry into account, the underrepresentation of AFR genetic ancestry was even starker. Cell lines characterized as AFR had large contributions from other (primarily European) ancestries; the average AFR genetic ancestry fraction for AFR cell lines was only 89%. In contrast, the average EUR and EAS ancestry fractions for EUR and EAS cell lines are 98.5% and 98.9%, respectively. Only four cell lines in this analysis were primarily SAS. No cell lines had greater than 80% AMR genetic ancestry, though AMR ancestry did comprise 16.3% of the genomes of Admixed cell lines. These imbalances in cell line ancestry limited statistical power to detect ancestry-associated dependencies among AMR and SAS cell lines. They also pointed out limitations to crude continental descriptors: dividing cell lines to binary ancestry groups without considering their local ancestry makeup would have resulted in the misclassification of all the admixed cell lines profiled in this analysis. Despite the stark imbalance across all continental ancestry groups, we did maintain sufficient statistical power to detect putative ancestry-associated dependencies associated with cell lines derived from patients of AFR, EAS, or EUR descent (Fig. 1c, Source Data 1C).

We next evaluated whether gene dependencies could be discovered that were significantly positively or negatively associated with a single local ancestry at the transcription start site of the gene in question. From the 16,384 genes profiled in this analysis, 49 gene dependencies appeared to be associated with either AFR (n = 24), EAS (n = 18), or EUR (n = 17) ancestry; surprisingly, we also detected gene dependencies that are associated with AMR (n = 3) and SAS (n = 1) ancestry, even though we lacked statistical power to reliably detect such associations (Fig. 1d, e, Source Data 1D, Source Data 1E). Many of these putative ancestry-dependency associations (13/49) had a reciprocal relationship with ancestry: each was both positively associated with either EUR or EAS ancestry and negatively associated with the other. This is likely because 91% of the cell lines included in our analysis are either European or East Asian. The current dataset is comprised of CRISPR screening data from two independently designed CRISPR libraries. To evaluate the impact of library design, we also performed a restricted analysis that included only cell lines screened with the Avana library (n = 558 cell lines) and discovered 46 dependencies putatively associated with ancestry, 34 of which were also discovered in the original pan-library analysis (Supplemental Fig. 1).

### Germline variants in ancestry-associated dependencies
We hypothesized that some of these associations between ancestry and dependencies were due to genetic differences in germline sequences. We therefore searched among SNP loci for dependency quantitative trait loci (hereafter referred to as d-QTLs) that could explain the putative differences in dependencies between ancestries. Specifically, we looked genome-wide to identify SNPs that are associated with each genetic dependency and unveiled the strongest association for such dependency, labeling such SNP as a putative d-QTL. We further prioritized SNPs as bona fide d-QTLs if they crossed genome-scale significance after correction for multiple hypothesis testing (FDR < 0.05). We detected 33 such associations across the 49 dependencies (Fig. 2a, Source Data 2AB). Among these SNPs, 25 (76%) were also associated with ancestry (Fig. 2b, Source Data 2AB), which we defined as having a differential minor allele frequency greater than 0.2 across two or more ancestry groups. In 29/49 cases the most significant SNPs for each gene were within 1 Mb from the transcription start site of the dependency gene (Fig. 2c, Source Data 2C), and SNPs that were within 1 Mb of the TSS had a stronger association with the dependency on the gene in question. Taken together, these data suggest that specific germline variants may influence a subset of ancestry-associated genetic dependencies.

Across The Cancer Dependency Map, gene expression has been previously observed to be the strongest predictor of gene

dependencies[8]. However, we found that the d-QTL SNP was associated with expression of the dependency gene in only 4/32 (15%) of cases (q < 0.05) (Fig. 2d, Source Data 2D). Furthermore, deeper evaluation of one putative positive association, specifically between chr9:21986219:C:T and *CDKN2B* expression unearthed an artifact related to SNP6 genotyping of samples with loss of heterozygosity at the *CDKN2B* locus (Supplementary Fig. 2). Similar artifacts were not observed with other genes. Thus, in aggregate, these data suggest that in most cases, d-QTLs are not modulating the expression levels of associated genes. Overall, for all ancestry-associated dependencies, 30/49 (61%) had expression levels below five reads per million, indicating that these genes are weakly expressed or not expressed in a majority of the profiled cell lines (Fig. 2e, Source Data 2E). The finding that so many genes that appeared to underlie ancestry-associated genetic dependencies were only weakly expressed further suggests that the variations in response to CRISPR/Cas9 targeting of many of these genes might reflect something other than true biological differences in gene dependency, such as a technical artifact.

### SNP mismatches in sgRNA targeting sequences
We therefore considered the possibility that many of these putative differences in cell line responses might be due to differences in the efficiency with which these sgRNAs were able to induce double strand breaks. Indeed, we found that across the 29 putative ancestry-associated dependencies with identified proximal d-QTLs, the d-QTLs for 11 (38%) were either germline variants in one or more of the sgRNAs targeting the relevant gene, or in linkage disequilibrium with such a variant. Only one of the putative ancestry-associated dependencies without an identified proximal d-QTL had such germline variants (Supplemental Fig. 3). Mismatches between a CRISPR/Cas9 sgRNA and the target genome preclude guide binding and subsequent genome editing in some circumstances[17,18], and the frequency of this variation can differ across ancestry groups[19,20]. CRISPR/Cas9-mediated double strand breaks negatively impact cell viability, and can lead to cell death independent of the genomic locus that is targeted by Cas9[21–24].

These observations support the hypothesis that variation between CRISPR/Cas9 guide and target sequences may explain a substantial fraction of putative genetic ancestry-associated dependency predictions. To comprehensively assess this, we deconstructed the consensus gene dependency scores[25], which aggregate signals across multiple sgRNAs, into 183 individual sgRNA scores across the 49 dependencies. We then tested the hypothesis that germline SNPs in targeting sequences influenced the differential effects between ancestry groups. As expected, we found that differences in sgRNA depletion between EAS and EUR cell lines was greater for guides with SNPs than for guides without SNPs ($p < 10^{-7}$, Fisher's exact test) (Fig. 3a, Source Data 3A). Indeed, this association extended past ancestry-associated variants. Across all sgRNAs in the Avana portion of the present dataset, 3209 (4.36%) have a SNV in their targeting sequence in at least 10 cell lines (Fig. 3b, Source Data 3B). Among these, 56% show a significant association between the presence of a variant and guide dependency (Supplementary Fig. 4). These guides account for 2.45% of all sgRNAs in The Cancer Dependency Map dataset.

Next, we hypothesized that the aforementioned analysis substantively undercounted the true magnitude of the artifact because SNP6 genotyping arrays do not detect all genetic variants, especially those that are very rare or are more specific for non-European ancestries. Indeed, in 297 cell lines, 25–30% of variants were detected exclusively by WGS but not by SNP6 genotyping arrays. Strikingly, cell lines from AFR individuals had more such variants missed by SNP6 than cell lines from any other ancestry group (Supplemental Figs. 5, 6). To understand the true magnitude of how many guides are affected, we leveraged WES/WGS variant calls to identify mismatches in targeting locations for Avana guides (see Data Availability). Across all

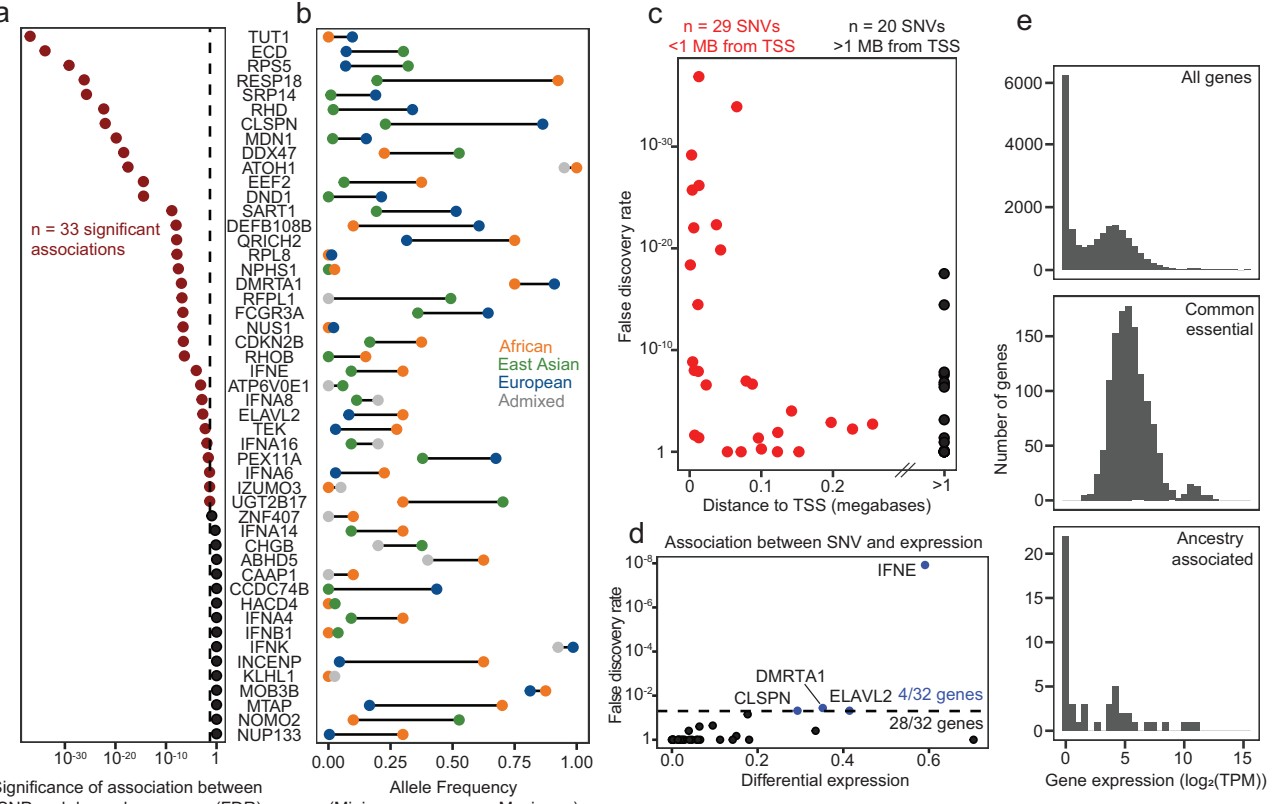

**Fig. 2 | Germline genetic variants underlie most putative ancestry-associated dependencies. a** The association between all genetic variants (from SNP6 genotyping arrays) and all putative ancestry-associated genetic dependencies was computed by linear regression with correction for cancer lineage as a covariate. Nominal *p*-values were adjusted for multiple hypothesis testing with the Benjamini–Yekutieli[43] procedure. Genes with at least one significant association (d-QTL) are indicated in red (*n* = 33). Raw data are described in Source Data 2AB. **b** The minor allele frequency of the most significant variant from **a** is indicated for the ancestry group with the lowest and highest frequency in the cell line collection. Cell lines of predominantly American or South Asian ancestry were not included in this analysis because the sample sizes for these ancestry groups are too low. Raw data are described in Source Data 2AB. **c** Position of the most significant germline variant from **a** relative to the Transcription Start Site (TSS) of the dependency gene in

question. Variants that are within 1 megabase of the TSS are indicated in red (*n* = 29) and those that are greater than 1 megabase or are on a different chromosome are black (*n* = 20). Raw data are described in Source Data 2C. **d** Association between each d-QTL Single Nucleotide Variant (SNV) and expression of the dependency gene in question. Genes in this analysis were filtered to include only those with at least one significant d-QTL association (indicated in red in **a**). *CDKN2B* was excluded from this analysis because the association between the d-QTL variant and *CDKN2B* expression resulted from a technical artifact (Supplementary Fig. 2). Raw data are described in Source Data 2D. **e** Median expression across cell lines for all genes (top), all genes classified as common essential in The Cancer Dependency Map 22Q1 release (middle), and all ancestry-associated genetic dependencies (bottom). Raw data are described in Source Data 2E.

genes profiled in The Cancer Dependency Map dataset 4918 have a mismatch (in at least one cell line) in one guide, 6178 have such mismatches in two guides, 3923 have such mismatches in three guides, and 1112 have such mismatches in all four guides (Fig. 3c, Source Data 3C). In aggregate, this artifact impacts 89% of genes targeted by Avana guides. Despite this concerning statistic, reassuringly, most CRISPR screening libraries contain multiple guides targeting each gene, blunting the overall impact of this experimental artifact if computational methods appropriately aggregate signals into multi-guide gene scores. However, residual artifactual signal may remain as we still detected gene-level associations (after computational gene-level signal aggregation across multiple guides) between SNPs and genetic dependencies, even when only one (out of four total guides) for the given gene was affected (Supplemental Fig. 7).

Single nucleotide mismatches in an sgRNA targeting sequence can prevent guide binding and the cutting activity of Cas9, with some positions on the sgRNA being less tolerant to mismatches than others[3,19]. These mismatches can result from both germline and somatic alterations, with germline variants being up to 1000× more frequent than somatic alterations (Fig. 3d, Source Data 3D). Mismatches in the sgRNA targeting sequence that are closer to the

protospacer adjacent motif (PAM) are less tolerated than those that are further[3]. We therefore hypothesized that SNP mismatches in sgRNA targeting sequences should impact guide dependency as a function of their distance from the PAM. To test this hypothesis, we compared the location of each SNP mismatch to the magnitude of the difference in dependency between cell lines with and without it. Indeed, mismatches closer to the PAM had a greater impact than those that were further. For example, mismatches in the position farthest from the PAM were not associated with guide dependency ($p = 0.15$), whereas mismatches in the position closest to the PAM were strongly associated with guide dependency ($p < 0.001$) (Fig. 3e, Source Data 3E). The impact of mismatches on guide depletion are also significantly correlated with the known impact of mismatches on guide cutting activity[3] (Supplementary Fig. 8). In almost all cases, mismatches were protective against the cutting activity of the guide, further suggesting that variation in the sgRNA targeting sequence precludes Cas9-mediated genome editing (Supplementary Fig. 4).

**Evaluation of sgRNA mismatches across ancestry groups**

We found ancestry-associated statistical bias in CRISPR guide design across all genetic ancestry groups, cell models, and CRISPR guide

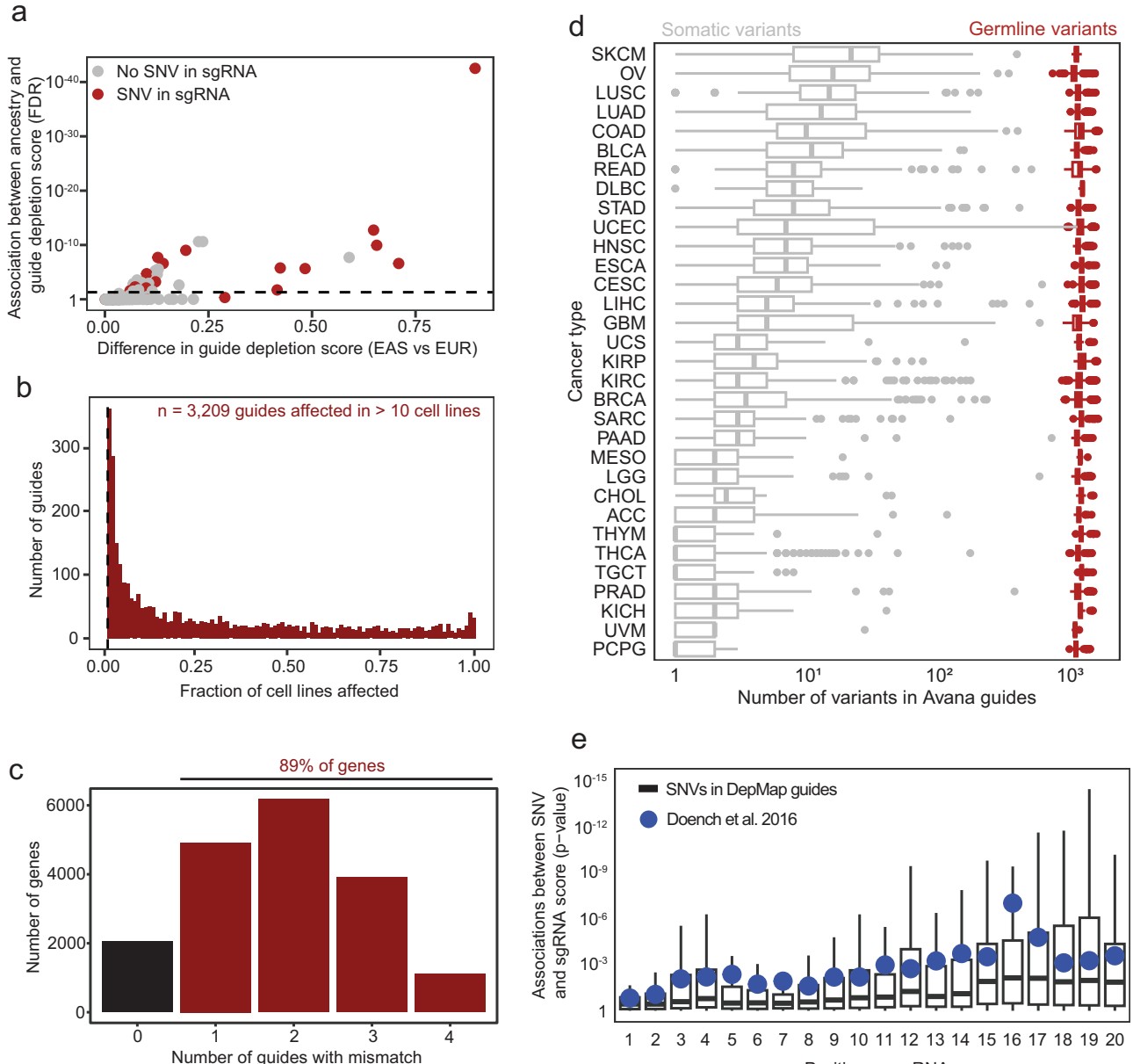

**Fig. 3 | Evaluation of ancestry-associated SNP mismatches in sgRNA targeting sequences. a** Association between East Asian (EAS) or European (EUR) ancestry and guide depletion scores for all guides targeting ancestry-associated dependencies. Guides with a Single Nucleotide Variant (SNV) in the targeting sequence are indicated in red. Raw data are described in Source Data 3A. **b** For all sgRNA sequences in the Avana library, the fraction of cell lines with a SNV in its targeting sequence. Guides which are affected in >10 cell lines (*n* = 3209) are indicated in red to the right of the dashed line, whereas those that are affected in <10 cell lines are not shown. Raw data are described in Source Data 3B. **c** Variants (from WES/WGS) for all cell lines were mapped to the targeting sequence of all Avana guides (see Data Availability). Genes are stratified by the number of guides with a mismatch in at least one cell line. Raw data are described in Source Data 3C. **d** Germline (red) and somatic (gray) variants from 32 tumor types profiled in TCGA were mapped to targeting

sequences for guides in the Avana library. The total number of variants in each sample that map to Avana guides is plotted on the x-axis. In the boxplot, the box includes the second and third data quartiles divided by a median line, and whiskers represent the first and fourth quartiles. Boxplot summary values are described in Source Data 3Db. Raw data are described in Source Data 3D. **e** Guides were stratified by the position of mismatches within the sgRNA targeting sequence and the association between the SNV and the guide depletion score was computed for each sgRNA in the Avana library (black boxes). *P*-values were computed with two-sided *t*-tests between cell lines with and without each SNV. The impact of mismatches on guide activity from Doench et al.[3] is indicated in blue circles. In the boxplot, the box includes the second and third data quartiles divided by a median line, and whiskers represent the first and fourth quartiles. Boxplot summary values are described in Source Data 3Eb. Raw data are described in Source Data 3E.

libraries that we evaluated. However, without explicitly accounting for ancestry effects, individuals of predominantly African descent are most affected because people of recent African descent are the most genetically diverse of any continent[26]. This is exemplified when sgRNAs are designed without considering human germline variation. To model this, we chose a random autosomal set of 1,000,000 loci with a canonical NGG PAM site and corresponding protospacer. We limited

the selection of these genomic loci to only those that are in protein-coding exons, since most CRISPR-based experiments target coding regions and genomic variability is lower in coding regions than non-coding regions. We collected SNP genotyping data from all 4120 gnomAD samples with individualized genotyping data, then mapped these SNPs to these regions. We found that 62.3% of these sgRNAs contained a targeting sequence SNP in at least one individual, and a

median of 1.80% of guides were affected in each individual (Fig. 4a, Source Data 4A). Individuals of African descent, however, were most affected by this artifact (2.17% in AFR, vs 1.78% in all other ancestry groups).

Multiple factors need to be optimized during the CRISPR sgRNA design process, including maximizing the likelihood that the guide will introduce the intended cut, minimizing the likelihood that the guide will introduce non-specific cuts at additional genomic loci, and minimizing the likelihood of mismatch between the sgRNA and its target due to human variation. For this latter factor, our results strongly suggest that differences in variant frequencies across populations should be accounted for, to ensure equal efficacy across populations.

Indeed, accounting for human variation in sgRNA design without explicitly accounting for differences in genetic variation across populations does not eliminate the statistical bias against individuals of African ancestry. We mapped germline variants from gnomAD to sgRNA targeting sequences from six genome-scale CRISPR libraries (Avana[3], Calabrese[27], Dolcetto[27], GeCKOv2[5,28], MinLibCas9[29], TKOv3[30], and HSANGERV[31]) (Fig. 4a, Source Data 4A). Among these seven libraries, five (Avana, Calabrese, Dolcetto, GeCKOv2, and MinLibCas9) were designed without attempting to avoid SNP loci in sgRNA targeting sequences. The other two (TKOv3 and HSANGERV) excluded sgRNAs targeting loci with a SNP listed in the db38 and Ensembl 1000 genomes databases, respectively. As might be expected, the five libraries that did not account for SNP variants had the greatest fraction of guides with mismatches due to human variation, especially in individuals of African descent (Fig. 4a, Source Data 4A). However, all libraries had higher mismatch rates in African individuals compared to other ancestry groups. Indeed, the ratio between failure rates in African individuals and other populations was surprisingly constant across all six libraries, ranging from 1.21 to 1.71 (Fig. 4b, Source Data 4B). The absolute failure rate was highest in the Calabrese and Dolcetto CRISPRi libraries (Fig. 4a, Source Data 4A), likely because these guides map to non-coding regions of the genome.

Although the absolute number of affected CRISPR guides in each individual is small (a median of 0.16–3.80% across the seven libraries), the impact of this artifact at the gene and cohort level can be large. The results above demonstrate that 89% of guides are impacted across the present dataset of 611 cell lines. When we evaluated the biological impact of this artifact, we found that in the Avana library, for example, 10–36 genes (median = 20) in the COSMIC Cancer Gene Census[32], are in fact affected by this artifact in each individual (Fig. 4c, Source Data 4C). Many such impacted genes, including *ACVR1*, *EGFR*, *TET2*, and *MET*, play important roles in cancer as oncogenes or tumor suppressors (Fig. 4d, Source Data 4D).

Accounting for ancestry-associated human genetic variation should drive changes in CRISPR library design. We sought to understand the impact of ancestry bias correction on CRISPR library design by performing in silico design of an improved genome-scale CRISPR/Cas9 library (herein referred to as Ancestry bias-corrected library), as described in Methods. Specifically, we avoided designing guides in regions with high levels of genomic variability in a general population or specifically in the African population (with a less strict threshold, see Methods). To benchmark how these restrictions affect guide quality, we performed two analyses. In the first analysis, we designed four guides for each gene and determined the number of genes with at least one guide targeting a variable genomic locus (Supplementary Fig. 9). In the second analysis, we computed the on-target score, a metric for the likelihood of on-target genome editing by a CRISPR sgRNA, for all genes and for genes with at least one guide targeting a variable genomic locus. Removing guides that have variants with a minor allele frequency of 0.1 (10% of the alleles in the population) resulted in a 1-2% decrease in overall guide quality (Supplementary Fig. 10). Overall, these analyses suggest that ancestry aware approaches to library design can be implemented without sacrificing reagent quality.

We corrected for this artifact in the CRISPR screening data in The Cancer Dependency Map to reduce the impact of sgRNA mismatches on the gene-level dependency scores (see Methods section). The corrected version of this dataset was included in the 22Q2 Dependency Map release. To understand the impact of this correction on the Dependency Map, we computed differential dependence for all genes pre- and post-correction. Interestingly, we identified 2223 genes with a significant difference in dependency scores following correction, although in almost all cases the magnitude of the difference is small (Supplementary Fig. 11). Using this corrected dataset, we revisited the original idea of identifying ancestry-associated gene dependencies. Excitingly, this analysis revealed 33 ancestry-associated dependencies after artifact correction (Fig. 4e, Source Data 4E) suggesting bona fide signals remain to be evaluated further.

## Discussion

The analytical pipeline used to compute Cancer Dependency Map gene dependency scores leverages a version of the Chronos[25] algorithm that we have now designed to be ancestry-aware. Correcting for genetic variation in sgRNA target sequences will likely be necessary for other similar algorithms including MAGeCK[33] and BAGEL[34]. Moreover, this artifact does not just affect large-scale CRISPR libraries; rather, it affects all CRISPR-based experiments. We have therefore also developed a web-based tool (www.ancestrygarden.org) that facilitates the discovery of sgRNA sequences that have high mismatch rates across ancestry groups both for the CRISPR libraries profiled in this study and for custom user-input sgRNA sequences.

We recognize that our classification schema for ancestry used herein has certain limitations. Specifically, while we use computational methods that derive continental ancestry groupings to highlight the importance of using diverse reference genomes for developing molecular tools, such continental labels can unintentionally conflate problematic uses of race and human genetic variation. A multi-dimensional and continuous conceptualization of ancestry can resolve some of these issues[35–37]. However, there is a lack of consensus on optimal ways to describe and visualize human genetic variation that are both precise and prevent harm to all people, including groups that have been negatively impacted by racist categorizations. Most individuals have mixed ancestries and global ancestries cross-classify continents[35–37]. Despite accounting for admixture in the present analysis, genomic loci were still classified into binary ancestries, undercounting diversity within ancestry groups. This is especially problematic for individuals of recent African descent, which in the present study we refer to as AFR, given the degree of genetic diversity in the African continent and diaspora. These genomic-defined ancestry groups are subject to evolutionary changes, as well ancestry-specific allele frequencies used to define these ancestry groups. These binary ancestry classifications will change as diversity in genetic reference databases improves and future evaluations of the sort presented here will be helpful.

Herein, we highlight a critical flaw in current CRISPR guide design practices, and we demonstrate the impact this artifact has on discovery of genetic dependencies in cancer. Earlier work also found that genetic variation within CRISPR/Cas9 sgRNA targeting sequences[19], particularly at therapeutically relevant loci[20], impacts sgRNA binding. However, until recently it was not possible to systematically understand the impact of this artifact on cancer target discovery. Furthermore, although the potential for this artifact was described over five years ago[19,20], most sgRNA design platforms still do not correct for it. With the present report, we hope to further raise awareness of this important issue and propose and implement a series of solutions to help the scientific community mitigate it.

We previously found that ancestry-associated artifacts can frequently arise in descriptive genomic data[38,39]; here we find that this also extends to functional genomic data. These findings highlight how

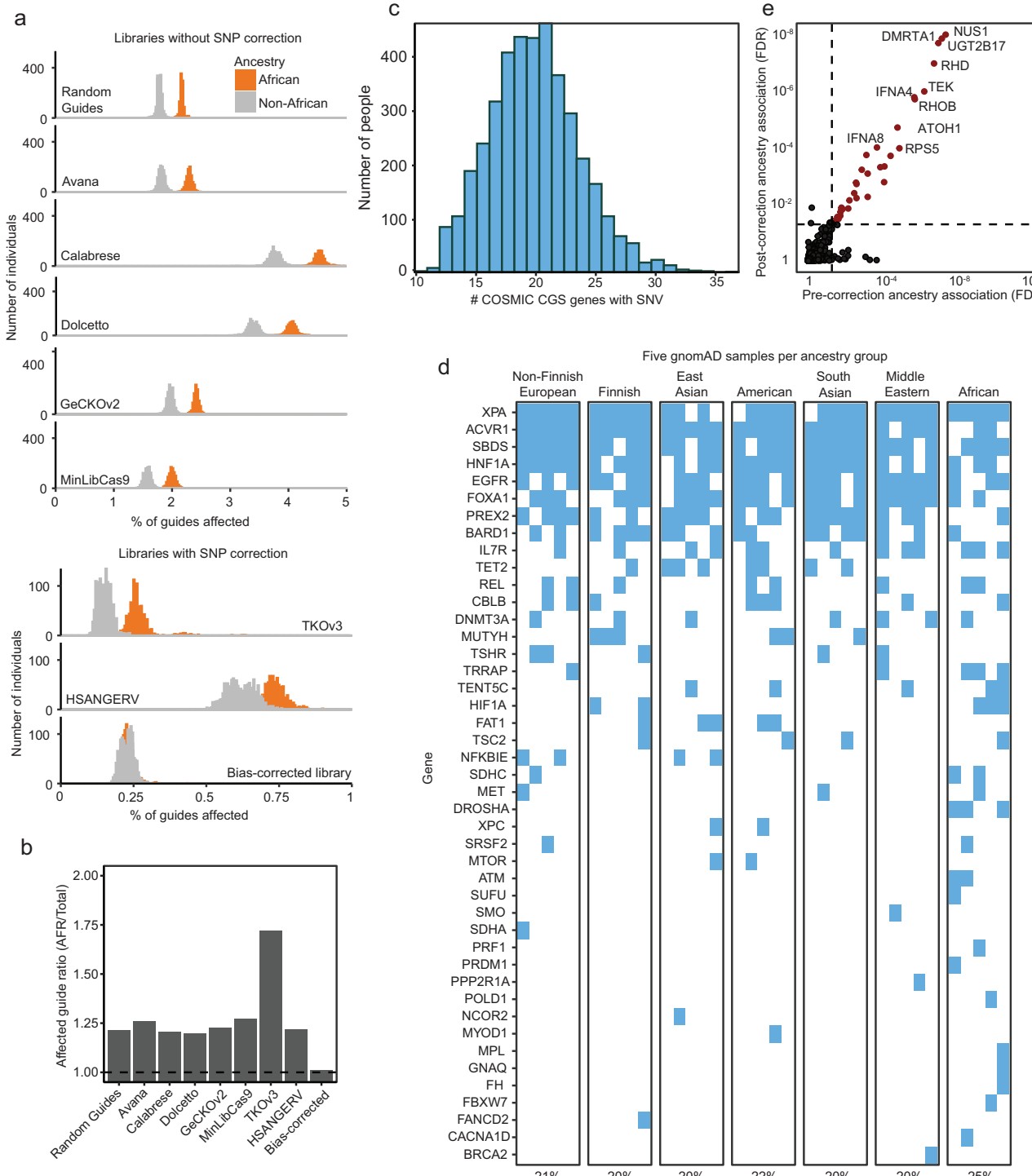

**Fig. 4 | CRISPR guide mismatches disproportionately affect individuals of recent African descent. a** The fraction of guides affected by genetic variants (horizontal axis) across individuals in the gnomAD dataset (vertical axis) for individuals of recent African descent (orange) or all other ancestry groups (gray). The gnomAD dataset was filtered to include only samples with individual-level genotypes and variants within each individual were mapped to sgRNA targeting sequences from 1,000,000 random guide-targeting loci (top), seven common genome-scale CRISPR libraries (middle), or a custom library designed as part of this study (bottom). Raw data are described in Source Data 4A. **b** The ratio between fraction of guides affected for African individuals compared to individuals from other ancestry groups from **a**. Raw data are described in Source Data 4B. **c** The number of Catalogue of Somatic Mutations in Cancer (COSMIC) Cancer Gene Census (CGC) Tier 1 genes affected in each individual (x-axis) by variants within

Avana guides for that gene, against the number of individuals so affected (vertical axis). Raw data are described in Source Data 4C. **d** Affected Avana guides for five randomly selected gnomAD samples from one of seven ancestry groups. Genes in which at least one guide has a mismatch in its targeting sequence are indicated with blue squares. Only genes annotated as oncogenes or tumor suppressor genes in the COSMIC CGS Tier 1 gene list were included. The numbers below the plot indicate the percentage of genes and samples affected within each ancestry group. Raw data are described in Source Data 4D. **e** Ancestry-associated dependencies (computed as described in Fig. 1d) pre- and post- correction (x- and y-axes, respectively) for variation within guide targeting sequences (see Methods). Red points are those that are statistically significant in both the pre-correction and post-correction datasets. Raw data are described in Source Data 4E.

widespread such ancestry-associated artifacts are across cancer research, often in ways that are invisible to researchers. The causes of cancer disparities are complex and multifactorial, but prejudicial biases in basic and pre-clinical research can form an important component. If we hope to make cancer outcomes equitable it is imperative that all forms of ancestry-associated statistical, experimental, and prejudicial biases are eliminated from cancer research.

## Methods

### Processing SNP6 genotyping calls

Publicly available SNP6 birdseed files[14] for 994 CCLE cell lines were converted to VCF files with BirdseedToVCF.py (see Code Availability section). The resulting VCF files for each individual sample were merged with bcftools (v1.16) to create a combined VCF file with all samples. The merged VCF file was split into 23 VCF files that contain variants from only a single autosome or chromosome X. Genotype calls were phased with Eagle (v2.4) and missing genotypes were imputed with Minimac4 (v1.6.6) using the TOPMed reference panel[40]. The resulting VCF files were merged to generate a single VCF file that contains all phased and imputed variants across all 994 cell lines.

### Local ancestry inference

Local ancestry was inferred for 994 CCLE cell lines with RFMix v2[41]. Imputed SNP6 variant calls for all 994 CCLE cell lines were filtered to include only variants that were detected in at least one of the 2504 unrelated samples profiled as part of the 1000 genomes project. Both the CCLE and the 1000 genomes datasets were segmented into identical genomic blocks with a minimum window size of 0.2 cM. Ancestry assignments were computed for each block and were assigned to one of five major ancestry groups (African, American, East Asian, European, or South Asian) using the 1000 genomes samples as a known reference for ancestry assignments.

To generate global ancestry assignment for cancer cell lines, the ancestry fraction within each chromosome was average across all chromosomes in each cell line. All chromosomes were equally weighted. Cell lines whose genomes are >80% of a single ancestry group were assigned as being predominantly that ancestry group, otherwise the cell line was assigned as being admixed.

### Identification of ancestry-associated genetic dependencies

We identified ancestry-associated genetic dependencies for all genes that were profiled in the Cancer Dependency Map dataset. First, for each individual cell line, we mapped the transcription start site of all genes to local ancestry blocks and assigned each gene to one of five major ancestry groups based upon the maximal ancestry fraction within that local ancestry block. For each gene, cell lines were binned by the ancestry assignment at that genetic locus. The association between genetic dependency scores and ancestry was computed by linear regression with inclusion of cancer lineage as a covariate (publicly available on the Cancer Dependency Map portal) and correction for multiple comparisons with the Benjamini-Hochberg False Discovery Rate method[42]. This analysis was restricted to only the 611 cell lines with both SNP6 genotyping and CRISPR screening data.

### Power calculations

For each cell line, the ancestry assignment was computed at the transcription start site of all genes profiled in The Cancer Dependency Map. Sample sizes $n = 26$ (AFR), $n = 6$ (AMR), $n = 203$ (EAS), $n = 373$ (EUR), $n = 4$ (SAS) were determined by computing, on average, the number of times in which a gene was assigned that ancestry group. Random distributions of Chronos scores, whose differences range from 0 to 1, in intervals of 0.01, were generated for all five ancestry groups. Statistical significance for one ancestry group vs all others was computed with linear regression with correction for cancer lineage.

This process was repeated 1000 times for each simulated difference in Chronos scores.

### Computing gene dependency scores

Gene dependency scores are routinely generated as part of ongoing bi-annual Cancer Dependency Map data releases. Quantitative gene-effect scores were computed using the combined datasets from the Broad Achilles (Avana library[3]) and Sanger SCORE (KY library[31])[25]. In the Avana library, 74,687 guides were tiled across 17,787 genes with 4 guides per gene. In the KY library, 101,094 guides (in the KY V1.1 library) were tiled across 17,349 genes with 5–10 guides per gene. Both libraries included a subset of guides which target intergenic regions. Guides for both libraries were independently designed[3,27]. Guide-level depletion scores underwent quality control and filtering steps[25] and lists of filtered and retained guides are included on the Cancer Dependency Map portal.

### Computing dependency QTLs (dQTLs)

Post-imputation SNP genotypes were filtered to only include those with minor allele frequencies >1% across all CCLE cell lines. Associations were computed between all SNPs and all ancestry-associated dependencies using a linear model with correction for cancer lineage (see Data Availability). Associations for each gene were corrected for multiple comparisons using the Benjamini–Yekutieli procedure[43]. SNPs with an FDR < 0.05 were considered to be statistically significant. For all genes with at least one significant dQTL, the variant with the lowest FDR was considered to be the marker dQTL for downstream analysis.

### Computing ancestry association for dQTLs

CCLE cell lines were subset to only include those in which genome-scale CRISPR screening has been performed, and those that are of predominantly African, East Asian, European, or Admixed ancestry. The allele frequency of each marker dQTL was computed for AFR, EAS, EUR, and Admixed cell lines individually, and across all cell lines.

### Identification of proximal and distal dQTLs

RefGene Transcription start site (TSS) positions for all genes were downloaded from the UCSC genome browser table viewer (genome.ucsc.edu). The distance between each dQTL and the TSS of the gene in question was computed. Distances less than one megabase were considered to be proximal, and distances greater than 1 megabase or those that are on a different chromosome were considered to be distal.

### Analysis of eQTLs

For each dQTL variant, CCLE cell lines were stratified into two categories, based upon presence (heterozygous alternate or homozygous alternate) or absence (homozygous reference) of each variant in question. The association between the genotype for each variant and gene expression (RNA-Seq) of the associated genetic dependency was computed using linear regression with correction for cell lineage as a covariate. This analysis was restricted to only include the subset of cell lines with both SNP6 genotyping and CRISPR screening data to maintain consistency with other analyses in this study.

### Mapping variants to sgRNA targeting sequence

This analysis was performed using both the subset of CCLE cell lines that were used to identify ancestry-associated dependencies and using the individual genotyping data for the HGDP + 1000-genomes call sets[44]. For analysis of CCLE cell lines, variants were mapped to the targeting sequences of all guides used in the Avana library. For analysis of the HGDP + 1000-genomes samples, variants were mapped to the targeting sequences of guides included in the Avana, Calabrese, Dolectto, GeCKOv2, MinLibCas9, TKOv3, and HSANGERV

libraries or to one million randomly selected sgRNA targeting sequences. See Data Availability statement for genomic loci of the targeting sequences for each library. Randomly selected sgRNA targeting sequences were selected by identifying all (NGG) PAM sites in coding regions. Guide targeting sequences were defined as the PAM site plus the 20 nucleotides on the 5' end of the PAM site. An identical method was used for mapping germline and somatic variants from 32 tumor types profiled in TCGA to targeting sequences of guides in the Avana library.

## Inferring positional impact of guide mismatches

The position of each mismatch was identified for all variants and all guides profiled in The Cancer Dependency Map. Across all guides, cell lines were binned into those with a mismatch in any guide (targeting any gene) in each position. The association between a mismatch in that guide position and dependence on the gene targeted by the guide in question was computed using a Wilcoxon test without correction for multiple hypothesis testing. These data were compared to ground truth data[3] wherein a library of guides with mismatches at each position were designed and screened to infer the importance of each position on guide fidelity.

## Correcting for ancestry bias in The Cancer Dependency Map

We first identified all mismatches between the targeting sequences of guides in the Avana library and the genomic sequences in each individual cell line. Guides with mismatches were excluded only for cell lines with a mismatch when calculating the gene-level dependency (Chronos) score. While this method will reduce the impact of mismatches on false negatives in CRISPR screens, one caveat is that the rate of impacted guides is higher in cell lines with AFR genetic ancestry. This results in a higher rate of eliminated guides in cell lines with AFR genetic ancestry than in cell lines from other genetic ancestry groups.

**Designing an ancestry-agnostic CRISPR library.** First, we leveraged the CRISPR sgRNA design tool CRISPick[3,27] to provide a ranked ordering of sgRNAs with the highest expected cutting rates for each gene while minimizing off-target effects. We then selected the four best sgRNAs for each gene that excluded SNPs with high frequencies (1% population frequency). We imposed the additional restriction that the mismatch frequency within guides may not be present at greater than a 2.5 times rate in AFR individuals than in non-AFR individuals. We found that these were the four CRISPick top-ranked sgRNAs for only 2222 (11.5%) of genes. This process resulted in similar rates of mismatches in African individuals (median 0.23%) as in individuals of other ancestry groups (also 0.23%) (Fig. 4a).

## Cell line sources

The cell line metadata source is described in the Data Availability statement. Cell line metadata was downloaded from the DepMap portal (depmap.org). We analyzed publicly available data from these cell lines as part of this study and did not perform cell line authentication. Cell lines which underwent DepMap profiling were authenticated, at the time of screening, with either STR profiling or SNP fingerprinting. The cell line source is listed in the 'SourceDetail' column, information for cell lines with ambiguous authentication results is listed in the 'PublicComments' column.

## Reporting summary

Further information on research design is available in the Nature Portfolio Reporting Summary linked to this article.

## Data availability

The input and intermediate data for figure generation are deposited on github (https://github.com/beroukhim-lab/ancestry_manuscript_code)

or on figshare (https://figshare.com/projects/Germline_variation_contributes_to_false_negatives_in_CRISPR-based_experiments_with_varying_burden_across_ancestries/202215). The DepMap data was accessed from the DepMap web portal (depmap.org) and download links are provided in the github README file (https://github.com/beroukhim-lab/ancestry_manuscript_code). The gnomAD (v3.1.2) datasets were downloaded from the gnomAD website (gnomad.broadinstitute.org). The CCLE SNP6 genotyping files were downloaded from[14]. DepMap WES/WGS data were downloaded from the DepMap web portal (depmap.org). The list of COSIC (v98) genes was downloaded from the COSMIC data portal (cancer.sanger.ac.uk.cosmic). TCGA somatic mutation MAF files were accessed from the GDC Data Portal (https://portal.gdc.cancer.gov) and germline mutations were accessed from a previously published study[45]. The publicly available CRISPR guide efficacy data from Doench et al.[3] are included as supplemental data in the associated manuscript (https://www.nature.com/articles/nbt.3437#Sec24). The publicly available CRISPR guide map data from Sanson et al.[27] are included as supplemental data in the associated manuscript (https://www.nature.com/articles/s41467-018-07901-8#Sec26). The publicly available data gnomAD genotyping data from Koenig et al.[44] are available on the gnomAD website (gnomad.broadinstitute.org). The publicly available TCGA germline variant data used in this study are available on the ISB cancer genome cloud and can be accessed with the following procedure (https://gdc.cancer.gov/about-data/publications/PanCanAtlas-Germline-AWG)[45]. Guide map matrices for Avana, Calabrese, Custom, Dolcetto, Gecko, Sanger, and TKO libraries are available on figshare (https://figshare.com/projects/Germline_variation_contributes_to_false_negatives_in_CRISPR-based_experiments_with_varying_burden_across_ancestries/202215). Cell line metadata for DepMap cell lines was downloaded from the DepMap data portal (see https://github.com/beroukhim-lab/ancestry_manuscript_code). The remaining source data are available within the Article, Supplementary Information, or Source Data file. Source data are provided with this paper.

## Code availability

All code used in this manuscript for data analysis or for figure generation are deposited on Github (https://github.com/beroukhim-lab/ancestry_manuscript_code).

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

## Acknowledgements

This work was supported by NIH grants: F32CA284834 (S.M.), R01 CA188228 (R.B.), R01 CA219943 (R.B.), R01 CA262462 (R.B.), R01 CA248280 (J.S.B.). This work was supported by DoD grants W81XWH-21-1-0901 (R.B.) and WX81XWH-21-1-0934 (J.S.B.). This work was supported by foundation grants from Alex's Lemonade Stand Foundation (S.M.), Pediatric Brain Tumor Foundation (R.B.), Gray Matters Brain Cancer Research Foundation (R.B.), and Innovations in Cancer Informatics (R.B.). This work was supported by the Bridge Project, a partnership between the Koch Institute for Integrative Cancer Research at MIT and the Dana-Farber/Harvard Cancer Center.

## Author contributions

S.A.M., R.B., and J.S.B. conceived the project. S.A.M. and A.F. carried out all data analysis. J.K. and J.N. carried out DepMap WES/WGS analysis. I.B. and J.D. carried out analysis of SNPs in DepMap guides under the guidance of J.M. A.F., L.P., and K.H. created AncestryGarden. A.F., P.R., T.G., and A.B. performed integration of mismatch frequencies into CRISPick and performed subsequent analysis under the guidance of J.D. and D.R. S.M., A.F., A.S., R.B., and J.B. drafted the manuscript. All co-authors reviewed and approved the final version of the manuscript.

## Competing interests

R.B. owns equity in and consults for Scorpion Therapeutics and receives grant funding from Novartis. All remaining authors declare no competing interests.
