## [Peer Review File · Nature Communications]

Germline variation contributes to false negatives in CRISPR-based experiments with varying burden across ancestriesEditorial Note: This manuscript has been previously reviewed at another journal that is not operating a transparent peer review scheme. This document only contains reviewer comments and rebuttal letters for versions considered at Nature Communications.

Reviewers' Comments:

Reviewer #2:

Remarks to the Author:

We appreciate the authors' thorough response to our comments. The revised manuscript clarifies the impact of somatic and germline variants in CRISPR guide libraries. The revised text also highlights the impact of these variants on the gene-level Dependency Map dataset. The analysis of ancestry dependencies, post-correction, and inclusion of tissue covariates is more robust, resulting in identifying 33 potentially interesting associations. While the authors acknowledge that the overall impact on the current dependency dataset is reduced by multiple guides targeting a gene, addressing ancestry bias in future CRISPR libraries will be important for the ongoing screening of samples from a more diverse and representative population.

Reviewer #3:

Remarks to the Author:

The authors were entirely receptive to our previous comments and concerns. Also, the authors did a thorough job changing the manuscript in response to our feedback to them. Overall, we are of the feeling that the modifications are satisfactory.

Reviewer #4:

Remarks to the Author:

Authors have responded comprehensively to the issues arising from the first version of this manuscript.